# Mitochondrial Fission and Fusion: Molecular Mechanisms, Biological Functions, and Related Disorders

**DOI:** 10.3390/membranes12090893

**Published:** 2022-09-16

**Authors:** Mode Al Ojaimi, Azza Salah, Ayman W. El-Hattab

**Affiliations:** 1College of Medicine, University of Sharjah, Sharjah 27272, United Arab Emirates; 2Pediatrics Department, University Hospital Sharjah, Sharjah 72772, United Arab Emirates; 3Genetics and Metabolic Department, KidsHeart Medical Center, Abu Dhabi 505193, United Arab Emirates

**Keywords:** mitochondrial diseases, mitochondrial fission, mitochondrial fusion, mitochondrial dynamics

## Abstract

Mitochondria are dynamic organelles that undergo fusion and fission. These active processes occur continuously and simultaneously and are mediated by nuclear-DNA-encoded proteins that act on mitochondrial membranes. The balance between fusion and fission determines the mitochondrial morphology and adapts it to the metabolic needs of the cells. Therefore, these two processes are crucial to optimize mitochondrial function and its bioenergetics abilities. Defects in mitochondrial proteins involved in fission and fusion due to pathogenic variants in the genes encoding them result in disruption of the equilibrium between fission and fusion, leading to a group of mitochondrial diseases termed disorders of mitochondrial dynamics. In this review, the molecular mechanisms and biological functions of mitochondrial fusion and fission are first discussed. Then, mitochondrial disorders caused by defects in fission and fusion are summarized, including disorders related to *MFN2, MSTO1, OPA1, YME1L1, FBXL4, DNM1L*, and *MFF* genes.

## 1. Introduction

Mitochondria are double-membrane organelles composed of a mitochondrial outer membrane (MOM) and a mitochondrial inner membrane (MIM) separated by the intermembrane space (IMS). The MIM is impermeable to most solutes, encloses the mitochondrial matrix, and forms cristae that expand its surface area. The electron transport chain (ETC) complexes are embedded in the MIM. They generate ATP via oxidative phosphorylation (OXPHOS), which involves electron transfer via complexes I-IV and ATP synthesis via complex V. Mitochondria are under dual genetic control with more than 99% of mitochondrial proteins are encoded by nuclear DNA (nDNA), while mitochondrial DNA (mtDNA) encodes less than 1% of mitochondrial proteins [1,2].

Mitochondria are dynamic organelles that constantly undergo fusion and fission, which provide mitochondria with a very interactive behavior [3]. Mitochondrial fission and fusion are active processes that occur continuously and simultaneously and are mediated by nDNA-encoded proteins that act on mitochondrial membranes. These specialized proteins include mechanical enzymes that alter the mitochondrial membranes physically and adaptor proteins that facilitate the binding of the mechanical enzymes to the organelles. Mitochondrial fission creates new mitochondria during cell division, allows the redistribution of mitochondria, and facilitates the segregation of damaged mitochondria, whereas mitochondrial fusion enables the exchange of intramitochondrial material between mitochondria. The balance between these two cellular processes determines the mitochondrial morphology and adapts it to the metabolic needs of the cells [4]. These two processes are crucial to optimize mitochondrial function and its bioenergetics abilities. Mitochondrial morphology reflects the respiratory activity of the cell. Maximum respiratory activity necessitates the fusion of the mitochondria, whereas, during cellular nutrient excess or cellular dysfunction, mitochondrial fragmentation occurs. In respiratory-active cells, mitochondria fuse to allow spreading of mitochondrial contents, counteract the effect of mitochondrial mutations that accumulate with aging, and optimize mitochondrial function. Mitochondria fragment in resting cells in order to remove damaged content by autophagy [5].

Defects in mitochondrial proteins involved in fission and fusion due to pathogenic variants in the genes coding them result in disruption of the equilibrium between fission and fusion, leading to a group of mitochondrial diseases termed disorders of mitochondrial dynamics [6]. Herein, the molecular mechanisms and biological functions of mitochondrial fusion and fission are first discussed; then, known mitochondrial disorders caused by defects in fission and fusion are summarized.

## 2. Mechanisms and Functions of Mitochondrial Fission and Fusion

Both fission and fusion are mediated by nDNA-encoded proteins that act on mitochondrial membranes (Figure 1). These include a small number of highly conserved guanosine triphosphatases (GTPases) proteins and their interactors, which regulate these opposite processes.

### 2.1. Mitochondrial Fusion

Mitochondrial fusion is a two-step process that involves the simultaneous fusion of the MOM and MIM. It is mediated by three dynamin-related GTPase: mitofusin 1 (MFN1; encoded by *MFN1*), mitofusin 2 (MFN2; encoded by the *MFN2*), and optic atrophy 1 protein (OPA1; encoded by *OPA1*) [7] in addition to two proteins: Misato (MSTO1; encoded by *MSTO1*) and F-box and leucine-rich repeat 4 (FBXL4; encoded by *FBXL4*) [8].

Mitofusins mediate MOM fusion. They are MOM proteins with two proposed topologies. In the initial one, there is a GTPase domain located at the N-terminus, one hydrophobic heptad repeat (HR1), the transmembrane anchor(s), and a second hydrophobic heptad repeat (HR2). Accordingly, both the N- and C-terminus are facing the cytosol, and there are two transmembrane domains [9,10]. In the second proposed topology, the C-terminus is not facing the cytosol and resides in the IMS, and there is only one single-spanning membrane domain [11]. The GTPase and tethering actions of MFN1 are bigger than those of MFN2; hence, MFN1 is believed to be the principal GTP-dependent membrane tethering protein for mitochondrial fusion [12]. Although MFN2 shares ~80% sequence identity with its homolog MFN1, a proline-rich region (PR domain) following HR1 is only present in MFN2, accounting for particular protein–protein interactions [13]. In fact, alteration in either one of them leads to a different inhibition of the fusion reaction [7]. Embryonic fibroblasts deficient in MFN1 or MFN2 show different forms of fragmented mitochondria. In cells missing MFN1, mitochondria fail to bind, indicating that MFN1 works in mitochondrial tethering, whereas MFN2 operates in a later process of the fusion reaction [13]. The stability and activity of MFN1 are regulated by acetylation and ubiquitination, while MFN2 can undergo ubiquitination only [14,15].

Misato (MSTO1; encoded by the *MSTO1* gene) protein is a soluble cytoplasmic protein that translocates to the MOM and interacts with the mitochondrial fusion proteins at the MOM–cytoplasm interface. MSTO1 can support mitochondrial fusion through enhancing or initiating the MOM fusion [8]. Its depletion causes mitochondrial fragmentation [16].

OPA1 is the main regulator of MIM fusion and cristae remodeling [17]. It is present in eight isoforms in humans and contains three highly conserved regions that are exposed to the IMS: the GTP-binding domain, the middle domain, and the GTP-effector domain. In addition, the N-terminus region includes a mitochondria-targeting sequence followed by a transmembrane helix that is needed for anchoring the MIM [18]. It directly links mitochondrial structure to bioenergetics function. When the transmembrane potential across the MIM is intact, long OPA1 (L-OPA1) isoforms carry out MIM fusion. When the potential is lost, L-OPA1 is cleaved to short (S-OPA1) by stress-sensitive IMS metalloendopeptidase OMA1 (OMA1; encoded by the *OMA1* gene) [19] with increased S-OPA1 inhibiting the fusion process and promoting mitochondrial fragmentation [20]. The ATP-dependent zinc metalloprotease YME1L1 (YME1L1; encoded by *YME1L1*) catalyzes the degradation of OMA1 in response to membrane depolarization [21]. This proteolytic mechanism is a regulator of organellar function and structure. It engages directly with apoptotic factors as the main mechanism for mitochondrial participation in the cellular response to stress.

F-Box and Leucine-rich repeat protein 4 (FBXL4; encoded by the *FBL4* gene) is a nuclear-encoded mitochondrial protein located in the IMS. Through its leucine-rich repeat domain, it can engage in protein–protein interactions, allowing it to form quaternary protein complexes. FBXL4 may play a role in mitochondrial fusion via interacting with and regulating mitochondrial fusion proteins [22].

Mitochondrial fusion allows the cells to build interconnected mitochondrial networks by producing tubular or elongated mitochondria. These networks act as united systems, promoting oxidative phosphorylation and leading to efficient dissipation of energy in the cells. Therefore, these networks are frequently found in metabolically active cells [23] Mitochondrial fusion also enables content mixing within the mitochondrial population, hence avoiding permanent loss of essential components and unifying the mitochondrial compartment, which optimizes mitochondrial function [24]. This mixing also allows the redistribution of mtDNA between damaged and healthy mitochondria, which enables human cells to tolerate high levels of pathogenic mtDNA and prevents mitochondrial elimination via mitophagy [25,26]. Hence, mitochondrial fusion is an essential regulatory process for optimizing mitochondrial function and enhancing mitochondrial integrity by allowing component sharing.

### 2.2. Mitochondrial Fission

MOM fission is a multistep process where a mitochondrion divides into two smaller mitochondria. It depends on a large cytoplasmic GTPase dynamin-1-like protein (DNM1L; encoded by *DNM1L)* that translocates to the MOM in response to cellular and mitochondrial signals. DNM1L acts on several MOM receptors, including mitochondrial fission factor (MFF; encoded by *MFF*), mitochondrial dynamics protein 49 (MID49), and mitochondrial dynamics protein 51 (MID51) [27]. When DNM1L is recruited to MOM, it forms a ring-like structure around the mitochondria, leading MOM constriction, which marks a potential site for future mechanical scission. Other mitochondrial fission sites are also marked by the endoplasmic reticulum and the actin cytoskeleton, which facilitates the oligomerization of the recruited DNM1L [28]. This is followed by GTP binding and hydrolysis, leading to a conformational change in DNM1L resulting in membrane scission. Post-translational phosphorylation, SUMOylation, and ubiquitination regulate DNM1L on the mitochondria [29].

Mitochondrial fission creates new mitochondria, which is crucial for rapidly dividing and growing cells to populate them with adequate numbers of mitochondria [30]. Mitochondrial fission also allows the redistribution of mitochondria. Through the formation of smaller mitochondria, mitochondrial fission allows a more efficient redistribution of these smaller sized organelles to the energy-demanding regions. Mitochondrial fission also contributes to quality control by enabling the removal of damaged mitochondria as it can isolate impaired mitochondria to be eliminated by mitophagy, which maintains mitochondrial homeostasis [26]. Therefore, mitochondrial fission is essential for mitochondrial distribution and homoeostasis.

## 3. Disorders of Mitochondrial Fission and Fusion

Impaired fusion results in fragmented mitochondria because of imbalanced fission, whereas defects in fission result in elongated mitochondria that are excessively connected because of unbalanced fusion [31,32]. Pathogenic variants in the genes coding proteins mediating fission and fusion result in the disruption of the equilibrium between fission and fusion leading, to impaired mitochondrial energy production. These mitochondrial diseases are called disorders of mitochondrial dynamics [6].

We hereby discuss the diseases of mitochondrial fusion that result from pathogenic variants in *MFN2* (Charcot–Marie–Tooth neuropathy 2A and hereditary motor and sensory neuropathy VIA with optic atrophy disease), *MSTO1* (mitochondrial myopathy and ataxia), *OPA-1* (optic atrophy 1, optic atrophy plus syndrome, Behr syndrome, mitochondrial DNA depletion syndrome 14), *YME1L1* (optic atrophy 11), *FBXL4* (mitochondrial DNA depletion syndrome 13), and the diseases of mitochondrial fission that result from pathogenic variants in *DNM1L* (encephalopathy due to defective mitochondrial and peroxisomal fission 1 and optic atrophy 5), and *MFF* (encephalopathy due to defective mitochondrial and peroxisomal fission 2) (Table 1).

### 3.1. MFN2-Related Disorders

Pathogenic variants in the *MFN2* (MIM*608507) cause three overlapping phenotypes: Charcot–Marie–Tooth neuropathy type 2A2A (CMT2A2A, MIM#609260), Charcot–Marie–Tooth neuropathy type 2A2B (CMT2A2B, MIM#617087), and hereditary motor and sensory neuropathy VIA with optic atrophy disease (HMSN VIA, MIM#601152).

Charcot–Marie–Tooth neuropathy type 2A (CMT2A) is the most common inherited axonal neuropathy characterized by distal muscle weakness and atrophy as well as sensory deficits [33]. In 90% of cases, CMT2A is caused by monoallelic pathogenic variants in *MFN2* (Charcot–Marie–Tooth type 2A2A) and is inherited as autosomal dominant, while 10% of cases occur due to biallelic pathogenic variants in *MFN2* (Charcot–Marie–Tooth type2A2B) and are inherited as autosomal recessive [34] or semi-dominant (i.e., a pathogenic variant is associated with mild disease in the heterozygous state and more severe disease in the homozygous or compound heterozygous state) [35]. A total of 25% of individuals with monoallelic pathogenic variants may be asymptomatic and have a normal electrophysiological examination which suggests incomplete penetrance [33]. The phenotype in those individuals could eventually convert to late-onset disease [36].

Age of onset ranges from 1 to 60 years with most of the autosomal recessive cases having early-onset disease (age < 10 years), which is associated with more severe disability than later onset. The initial presenting sign is mainly foot weakness or foot drop. Involvement of the lower extremities is more severe and seen earlier than the upper extremities, which become involved later in the course of the disease. Affected individuals have motor deficits (limping gait, difficulty running, difficulty climbing stairs, postural tremor, and distal muscle weakness and atrophy), which are more prominent than the sensory ones (decreased sensation of pain and vibration in feet). Other neurologic manifestations include lower limb hyporeflexia or areflexia, pyramidal signs (extensor plantar responses, mild increases in muscle tone, preserved or increased reflexes), vasomotor dysfunction, and ocular anomalies (optic atrophy in 7% of autosomal dominant form and pale optic discs in 20% of autosomal recessive form) [37]. Approximately 60% of individuals with early-onset disease develop subacute optic atrophy with consequent slow recovery [38]. Rare findings include hydrocephalus, fatal subacute encephalopathy (vomiting, nystagmus, chorea, clouded consciousness, and dysautonomia), spasticity, sensorineural hearing loss, dysarthria, migraine, and early-onset stroke [39]. Vocal cord palsy with dysphonia, respiratory insufficiency, and skeletal anomalies (scoliosis, kyphosis, contractures, and hammertoes) have also been reported [40]. CMT2A has a progressive course. Nearly 27% of individuals become dependent on a wheelchair [34].

Median nerve motor conduction studies range from normal to slightly reduced. Nerve biopsy, which was previously the key diagnostic step, is being replaced by genetic testing, but it is still important in atypical cases [41]. Findings on nerve biopsy include a loss of large myelinated fibers with no myelin abnormalities, mitochondrial abnormalities, and, although rarely, presence of onion bulb structures. Electromyography studies show chronic denervation signs in more than 90% of cases. Neuroimaging abnormalities can show a defect in mitochondrial energy metabolism in the occipital cortex on magnetic resonance spectroscopy and periventricular/subcortical white matter lesions [37]. Muscle imaging may show intramuscular fat accumulation, which may be associated with functional outcomes. Diagnosis is confirmed molecularly by identifying pathogenic monoallelic or biallelic variants in *MFN2*.

Hereditary motor and sensory neuropathy VIA with optic atrophy disease (HMSN VIA; Charcot–Marie–Tooth disease type 6A; CMT6A) is an autosomal dominant disease caused by monoallelic pathogenic variants in *MFN2* [38]. It is characterized by sensorimotor neuropathy and optic atrophy. Less than 100 individuals have been diagnosed to date. Peripheral neuropathy is early-onset (childhood to mid-adulthood; typically, between 10 and 30 years of age) with later onset of optic atrophy (mean 19 years, range 5 to 50 years), which frequently leads to visual loss. Neurological symptoms include loss of motor skills, hypertonia, hyper/hypo/areflexia, and ataxia. Reported ocular anomalies include optic atrophy, central scotoma, dysmetric saccades, pale optic disks, subacute deterioration of visual acuity, color vision defects, abnormal visual-evoked potentials, cogwheel ocular pursuit, profound visual loss with rod–cone dysfunction, extropia, nystagmus, and cataract. Neuromuscular symptoms include sensorimotor axonal neuropathy and proximal and distal muscle weakness with atrophy. Additional manifestations include cognitive impairment (cognitive decline, delayed motor and language development, and decreased IQ), sensorineural hearing loss, tinnitus, anosmia, vocal cord paresis, myalgia, and musculoskeletal anomalies (steppage gait, scoliosis, lumbar hyperlordosis, pes cavus, major joint contractures, and foot deformities). Serum creatine phosphokinase (CPK) and lactate may be elevated. Neuroimaging studies may reveal involvement of the periventricular white matter rather than the cerebral cortex [42], diffuse brain and cerebellar atrophy with cerebellar white matter abnormalities, calcifications in the basal ganglia, and chiasm atrophy [43]. Diagnosis is confirmed molecularly by identifying pathogenic monoallelic variants in the *MFN2* gene.

Management of *MFN2*-related disorders involves a multidisciplinary team that includes a neurologist, orthopedic surgeon, psychiatrist, and physical and occupational therapists. Standard medical treatment is supportive and based upon the affected individual’s needs. Routine visual assessment should be performed in individuals with or without optic atrophy and annually in children for educational needs. MRI of the legs to assess amount and location of fat replacing muscles should be performed every few years in specialized centers only [44]. Obesity, which makes walking more difficult and neurotoxic medications (e.g., vincristine and taxols) should be avoided in affected individuals. Orthotics such as ankle foot orthoses are key in the rehabilitative approach since they improve walking velocity, balance, and ankle range of motion. Musculoskeletal pain can be alleviated with acetaminophen or nonsteroidal anti-inflammatory agents. Tricyclic antidepressants, carbamazepine, or gabapentin may decrease neuropathic pain.

Mitofusin agonists and activators are showing promising results as a therapeutic approach for CMT2A and other diseases of impaired neuronal mitochondrial dynamics [45,46]. Mitofusin agonists stabilize the fusion-permissive open confirmation of endogenous normal MFN1 or MFN2. This overcomes the dominant suppression of mitochondrial fusion induced by the dysfunctional proteins and directly stimulates mitochondrial fusion in order to restore the balance between mitochondrial fission and fusion [45,47]. In mice that express human MFN2 T105M, intermittent activation of mitofusin using MiM111, which is a metabolically stable mitofusin activator with good nervous system bioavailability, normalized CMT2A neuromuscular dysfunction secondary to accelerated primary axonal outgrowth and greater postaxotomy regrowth [46]. Gene therapy is another promising approach. It has been shown that in vivo augmentation of MFN1 in the central nervous system of mice, using a transgenic approach, rescued all phenotypes in mutant MFN2-expressing mice [48].

### 3.2. MSTO1-Related Mitochondrial Myopathy and Ataxia

*MSTO1*-related mitochondrial myopathy and ataxia (MIM#617675) is caused by monoallelic pathogenic variants in *MSTO1* (MIM*617619) leading to an autosomal dominant disease or biallelic variants in *MSTO1* leading to a recessive disease. To date, a total of 27 cases from 19 families have been reported [49].

Age of onset is variable but mostly in early childhood. Common presenting features of both autosomal recessive and dominant diseases include intellectual disability, delayed motor development, learning disability, delayed speech, hearing impairment, ataxia with dysmetria and dysdiadochokinesis, hypotonia, tremor, difficulty walking, myalgia, muscle weakness and atrophy, short stature, distinctive facial features (small eyes, close-set eyes, micrognathia, prominent jaw, long face, and myopathic face), and musculoskeletal anomalies (scoliosis, delayed skeletal maturation, joint hyperlaxity, pes cavus, pes varus, and chest asymmetry). Features seen mainly in autosomal dominant variants include behavioral and psychiatric manifestations (anxiety, depression, and schizophrenia), endocrine abnormalities (delayed bone age, hyperthyroidism, hyperprolactinemia, and primary amenorrhea), lipomas, and frontal lobe atrophy [8]. Common features in autosomal recessive cases include poor growth, papillary pallor, hyporeflexia, brain imaging abnormalities (cerebellar hypotrophy, hyperintense white matter abnormalities), distinctive facial features (thick hair and high arched palate), pectus excavatum, and increased serum creatinine kinase [50]. Muscle biopsy shows myopathic features with increased fiber size variation, increased number of abnormal mitochondria, mitochondrial degeneration, and decreased mitochondrial mtDNA content. Fibroblasts from affected individuals display fragmented mitochondria, mtDNA depletion, enlarged lysosomal vacuoles, and reduced nucleoids number [51].

### 3.3. OPA1-Related Disorders

Monoallelic pathogenic variants in *OPA1* (MIM*605290) are associated with two phenotypes: optic atrophy 1 (MIM#165500) and optic atrophy plus syndrome (MIM#125250), while biallelic pathogenic variants are associated with two other phenotypes: Behr syndrome (MIM#210000) and mitochondrial DNA depletion syndrome 14 (MIM#616896) [52,53].

Optic atrophy 1 is an autosomal dominant disorder, although recent studies have suggested semi-dominant inheritance [54]. It is characterized by childhood-onset bilateral vision loss, visual field defects, and optic nerve pallor. Its prevalence is 1:12,000–50,000, which makes it the most common inherited optic neuropathy if glaucoma is excluded [55]. Its penetrance is 43 to 100%. Most cases have an affected parent, but de novo pathogenic variants have been reported. Affected individuals are usually detected during vision screening at school in the first decade of life (median 5 years), but later onset (21–30 years) has been reported. Visual impairment is typically bilateral and symmetrical. It ranges from mild to severe (usually moderate with a visual acuity of 20/80 to 20/120). Legal blindness is rare. Visual loss is usually irreversible and progressive during puberty until adulthood, with very slow chronic progression subsequently [56]. Reported visual field defects are paracentral, central, and centrocecal. Although color vision defects in the blue–yellow or red–green axes are commonly reported [57], over 80% of cases have a mixed-color deficit [58]. Other reported ocular anomalies include strabismus (10%), horizontal nystagmus (5%), ptosis, and progressive external ophthalmoplegia from the third decade of life onwards.

Typical ophthalmologic examination findings include bilateral and symmetrical optic nerve pallor (cardinal sign, temporal, global) with a wedge-like papillary excavation. Optic nerve heads of cases are usually smaller than in age-matched controls [9]. Optical coherence tomography reveals loss of retinal nerve fiber thickness mostly evident in the temporal quadrant with relative sparing of the nasal quadrant [59]. Electrophysiology abnormalities include absent or delayed visual evoked potentials. Histological examination can reveal diffuse atrophy of the retinal ganglion cell layer associated with atrophy and loss of myelin within the optic nerve but without atrophy of the outer retinal layers. Collagen is increased and neurofibrils and myelin sheaths are decreased in the optic nerves, optic chiasm, and optic tracts [60].

Optic atrophy plus syndrome occurs in up to 20% of optic atrophy cases that have additional extraocular neurological complications. Sensorineural deafness is a prominent manifestation that is bilateral and begins in late childhood or early adulthood but may be congenital or subclinical. Other manifestations include adult-onset cerebellar or sensory ataxia (29%), axonal sensorimotor peripheral neuropathy (29%), exercise intolerance, myalgia, muscle weakness, and proximal myopathy (35%) [61]. Rare clinical presentations include spastic paraparesis mimicking hereditary spastic paraplegia, multiple-sclerosis-like illness, and hypotonia with dysphagia and gastrointestinal dysmotility [39,61].

Behr syndrome is a genetically heterogeneous disorder characterized by childhood-onset optic atrophy with ataxia and pyramidal signs (spasticity, weakness, and hyperreflexia). Posterior column sensory loss and intellectual disability may be present. Gradual gait difficulties develop in the second decade of life. Other reported findings include cerebellar signs (dysmetria, dysdiadochokinesis, and nystagmus), hypotonia, delayed development, hearing loss, dysarthria, musculoskeletal anomalies (pes cavus and severe contractures of the lower extremities), gastrointestinal anomalies (dysphagia, vomiting episodes, intestinal dysmotility, and severe constipation). Reported brain MRI anomalies include cerebellar atrophy, vermian atrophy, atrophy of optic nerves and chiasm, and mild periventricular leukomalacia [62]. Adult-onset disease, including optic atrophy and ataxia, has been rarely reported [62].

Mitochondrial DNA depletion syndrome 14 (encephalocardiomyopathic-type) is characterized by severe lethal infantile mitochondrial encephalomyopathy and hypertrophic cardiomyopathy. It was reported in two sisters who showed profound neurodevelopmental delay, hypotonia, peripheral hypertonia (opisthotonic posturing, from birth), feeding difficulties, and hypertrophic progressive cardiomyopathy. One sister had abnormal eye pursuits with a weak cry, while the other had sensorineural deafness, optic atrophy, and increased serum and cerebrospinal fluid (CSF) lactate. They died at the age of 10 and 11 months. Electron microscopy from one sister showed incomplete fusion of the MIM along with large mitochondria. Significant mtDNA depletion was found in the muscle biopsies from both sisters [63].

Diagnosis of OPA1-related disorders is confirmed molecularly in suspected cases with the above clinical features by identifying biallelic or monoallelic pathogenic variants in *OPA1.* Red ragged fibers (RRFs), mtDNA deletions, and COX-deficient fibers may be observed in skeletal muscles.

Treatment is supportive and targeted to the individual’s needs. Proposed treatments under investigation include genetic therapy to correct the mutation-induced splice defect [64], antioxidants (vitamin E, superoxide) [65], and idebenone drug therapy [66]. In 74 of 87 individuals with dominant optic atrophy, an increased visual acuity was observed after at least 7 months of administration of idebenone. Tolfenamic acid trial therapy has shown positive effects on mtDNA stability and amelioration of the energetic functions and the mitochondrial network morphology, depending on the type of OPA1 mutation [67]. Recent studies have shown promising results with mesenchymal stem cell therapy using human embryonic stem cells and induced pluripotent stem cells [68,69].

### 3.4. YME1L1-Related Optic Atrophy 11

Optic atrophy 11 (MIM#617302) is an autosomal recessive disease caused by biallelic pathogenic variants in *YME1L1* (MIM*607472) and is characterized by delayed psychomotor development, optic atrophy, and leukoencephalopathy. To date, this disease has been reported in four siblings who presented with intellectual disability, developmental delay, hearing impairment, optic anomalies (optic nerve atrophy with visual impairment), and leukoencephalopathy observed on brain MRI. Inconsistent features include ataxia, hyperkinesia, athetotic and stereotypic movements, macro-/microcephaly, and elevated lactate levels in blood and CSF [70]. Muscle biopsy revealed neurogenic changes (grouped fibers indicating denervation) and mitochondria with altered cristae morphology and paracristalline inclusions. Cultured fibroblasts showed an increase in fragmented and shortened mitochondrial networks, which is consistent with mitochondrial network fragmentation [70].

### 3.5. FBXL4-Related Mitochondrial DNA Depletion Syndrome 13

Mitochondrial DNA depletion syndrome 13 (encephalomyopathic-type) (MIM#615471) is a multisystem disorder caused by biallelic pathogenic variants in *FBXL4* (MIM*605654) and is characterized by congenital or early-onset encephalopathy, developmental delay, hypotonia, and lactic acidosis [71]. Less than 100 cases have been reported to date.

Affected individuals present with early infantile onset of encephalopathy, hypotonia, global severe developmental delay, seizures, ataxia, movement abnormalities (dystonia and choreoathetosis), microcephaly, distinctive facial features (narrow face, thick eyebrows, epicanthus, upslanting palpebral fissures, long eyelashes, synophrys, broad nasal bridge and tip, saddle nose, long and smooth philtrum, malformed ears, protruding ears, low-set ears, and everted lower lip vermillion), gastroesophageal reflux, hypospadias, arrhythmias, neuroimaging anomalies (delayed myelination, thin corpus callosum, leukodystrophy, cerebral atrophy, white matter lesions in the brainstem and basal ganglia, and arachnoid cysts), and metabolic derangements (increased serum lactate, alanine, and ammonia) [72]. Other features can include plagiocephaly, nystagmus, cataracts, neutropenia, recurrent infections, renal tubular acidosis, hypertrophic cardiomyopathy, scoliosis, small feet, and abnormal liver enzymes. Mitochondrial hyperfragmentation can be observed in cultured fibroblasts. Decreased mtDNA content and reduced activity of multiple ETC complexes can be observed in skeletal muscle and fibroblasts [73]. Survival varies with a median age of reported deaths of two years (range 2 days–75 months), although survival up to 36 years old has been reported.

Treatment is supportive. Sodium pyruvate was shown to improve the muscle strength and the quality of life of an infant with myopathic mitochondrial DNA depletion syndrome [74]. Recent studies have shown that mitochondrial deletion syndromes are often associated with secondary CoQ deficiency [75]. While some studies have advised assessment of muscle CoQ status in affected individuals in order to consider early CoQ supplementation as a candidate therapy [76], other studies have concluded that the use of CoQ therapy in mitochondrial deletion disorders is not efficacious [76]. Further studies are required to sort out this controversy.

### 3.6. DNM1L-Related Disorders

Pathogenic variants in *DNM1L* (MIM*603850) are responsible for two distinct phenotypes: encephalopathy due to defective mitochondrial and peroxisomal fission 1 (MIM#614388) and optic atrophy 5 (MIM#610708). *DNM1L*-related encephalopathy due to defective mitochondrial and peroxisomal fission 1 is a lethal childhood encephalopathy characterized by delayed psychomotor development and hypotonia. It can be caused by monoallelic pathogenic variants in *DNM1L* leading to a dominant disease or biallelic pathogenic variants in *DNM1L* leading to a recessive disease. Variants in dominant disease occurred de novo [77]. To date, 11 cases have been identified: 7 with the autosomal dominant disease and 4 with the autosomal recessive disease. Optic atrophy 5 has been reported in three unrelated French families.

Affected individuals with encephalopathy due to defective mitochondrial and peroxisomal fission 1 present in early infancy or childhood with neurologic regression, severe developmental delay, and hypotonia. Inconsistent findings include refractory epilepsy (clonic, focal, generalized tonic–clonic, and status epilepticus), cognitive decline, insensitivity to pain, decreased visual tracking, difficulty walking, areflexia, absent response to light stimulation, myoclonus, hypertonia, dysphagia, failure to thrive, respiratory insufficiency, microcephaly, distinctive facial features (pointed chin and deep-seated eyes), cardiomyopathy, and skeletal anomalies (broad thumbs and big toes) [78,79,80]. Increased serum and CSF lactate can be seen in some affected individuals. Brain MRI findings range from normal to progressive cerebral atrophy, demyelination, delayed myelination, thinning of the corpus callosum, T2-weighted hyperintense lesions in the cortex, and abnormal gyral pattern in frontal lobes [81]. Cultured fibroblasts show decreased elongated peroxisomes and tubular mitochondria with defects in mitochondrial and peroxisomal fission. Postmortem evaluation of one patient revealed mitochondrial cardiomyopathy characterized by abnormal cardiac myocytes with enlarged mitochondria [82].

*DNM1L*-related optic atrophy 5 is characterized by slowly progressive visual loss with variable onset from the first to third decades. Additional ocular abnormalities may include dyschromatopsia (blue–yellow), central scotoma, a slow decrease in visual acuity, and optic nerve atrophy. Mitochondria in mutant cells showed a highly elongated, tabulated, and hyperfilamentous network supporting an impairment of mitochondrial fission [83].

Treatment is supportive. Bezafibrate has shown promising results as far as improving mitochondrial fission and function in DNM1L-deficient cells [84]. It is an agonist of peroxisome-proliferator-activated receptor alpha, which is a ligand-activated transcription factor that increases the expression of multiple genes including nuclear-encoded respiratory chain genes [85]. Bezafibrate normalized growth, ATP production, and oxygen consumption in fibroblasts from affected individuals [84].

### 3.7. MFF-Related Encephalopathy

Encephalopathy due to defective mitochondrial and peroxisomal fission 2 (MIM#617086) is an autosomal recessive disease caused by biallelic pathogenic variants in *MFF* (MIM*614785). To date, six affected individuals have been reported. Affected individuals present with severe hypotonia, delayed psychomotor development, microcephaly, and abnormal signals in the basal ganglia. Inconsistent features include early-onset seizures, including hypsarrhythmia, optic atrophy, peripheral neuropathy, hyperreflexia, spasticity, swallowing difficulties, ocular anomalies (vision loss, pale optic discs, absent visual fixation, and high refractive errors), hearing loss, short stature, failure to thrive, and diffuse cerebellar atrophy [86,87]. Serum lactate may be normal or increased. Cultured fibroblasts can show elongated peroxisomes and mitochondria, suggesting a fission defect [86,87].

## 4. Summary

Mitochondrial fusion and fission are crucial for the maintenance of normal mitochondrial morphology and functions. Mitochondrial fission creates new mitochondria during cell division, allows the redistribution of mitochondria, and facilitates the segregation of damaged mitochondria. This process is mediated by several nDNA-encoded proteins that act on tethering the MIM and MOM. Defects in mitochondrial fusion have been associated with pathogenic variants in *MFN2, MSTO1, OPA1, YME1L1*, and *FBXL4.* Mitochondrial fusion enables the exchange of intramitochondrial material between mitochondria. Mitochondrial scission is mediated by DNM1L, and defects in mitochondrial fission have been associated with pathogenic variants in *DNM1L* and *MFF* genes. Defects in mitochondrial fusion and fission result in diseases with variable neurological defects, which accentuates the importance of balanced mitochondrial fission and fusion in neuronal function [88].

## Figures and Tables

**Figure 1 membranes-12-00893-f001:**
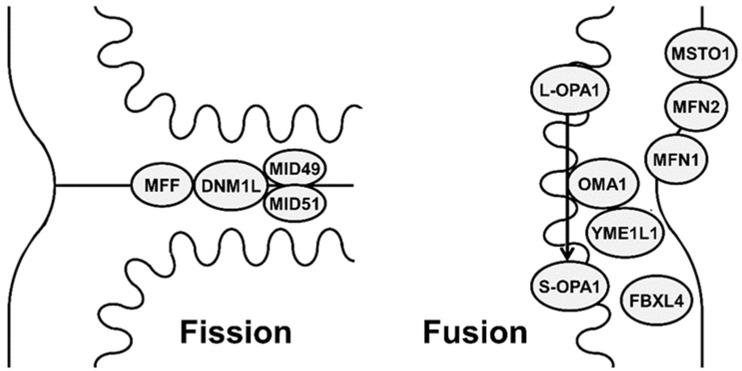
Schematic representation showing different mitochondrial proteins involved in mitochondrial fission (DNM1L (dynamin-1-like protein), MFF (mitochondrial fission factor), MID49 (mitochondrial dynamics protein 49), MID51 (mitochondrial dynamics protein 51)) and fusion (MFN1 (mitofusin 1), MFN2 (mitofusin 2), OPA1 (optic atrophy 1), MSTO1 (Misato), FBXL4 (F-box and leucine-rich repeat 4), metalloendopeptidase OMA1, and metalloprotease YME1L1).

**Table 1 membranes-12-00893-t001:** Disorders of mitochondrial fission and fusion.

Diseases		Gene	Main Clinical Manifestations
Mitochondrial fusion disorders	Charcot–Marie–Tooth neuropathy 2A	*MFN2*	Axonal sensorimotor neuropathy
	Hereditary motor and sensory neuropathy VIA with optic atrophy disease	*MFN2*	Axonal sensorimotor neuropathy and optic atrophy
	Mitochondrial myopathy and ataxia	*MSTO1*	Cognitive impairment, myopathy, and ataxia
	Optic atrophy 1	*OPA1*	Optic atrophy
	Optic atrophy plus syndrome	*OPA1*	Optic atrophy, hearing impairment, ataxia, neuropathy, and myopathy
	Behr syndrome	*OPA1*	Optic atrophy, ataxia, and pyramidal signs
	Mitochondrial DNA depletion syndrome 14	*OPA1*	Profound developmental delay, hypotonia, and hypertrophic cardiomyopathy
	Optic atrophy 11	*YME1L1*	Optic atrophy, developmental delay, and leukoencephalopathy
	Mitochondrial DNA depletion syndrome 13	*FBXL4*	Developmental delay, hypotonia, seizures, and lactic acidosis
Mitochondrial fission disorders	Encephalopathy due to defective mitochondrial and peroxisomal fission 1	*DNM1L*	Developmental delay, regression, hypotonia, and seizures
	Optic atrophy 5	*DNM1L*	Optic atrophy
	Encephalopathy due to defective mitochondrial and peroxisomal fission 2	*MFF*	Developmental delay, hypotonia, and microcephaly

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
