# Peer review of "Mitochondrial Fission and Fusion: Molecular Mechanisms, Biological Functions, and Related Disorders"

_membranes, 2022, doi:10.3390/membranes12090893_

Round 1

Reviewer 1 Report

Membranes 

COMMENTS TO THE EDITORS AND THE AUTHORS

membranes-1892615 “Mitochondrial fission and fusion: molecular mechanisms, biological functions, and related disorders”

Dear the Editor and the Authors,

Please find enclosed the comments for the above-mentioned manuscript.

A SUMMARY OF THE CONTENT

The authors stated that the study aimed to describe the molecular mechanisms and biological functions of mitochondrial fusion and fission as well as mitochondrial disorders caused by defects in fission and fusion including disorders related to MFN2, MSTO1, OPA1, YME1L1, FBXL4, DNM1L, and MFF genes.

THE OVERALL OPINION OF THE MANUSCRIPT

The strengths. The manuscript is within the scope of the journal.

The limitations. The main concern is credibility of the authors to write the review with the subject they never published. According to the PubMed, the authors did not publish any original article on the subject they intend to describe. These facts are quite visible through the text. Last author published few review articles on similar subject. There is significant amount of the similarities between reviews published this year and previously and this manuscript. In the abstract, 6 out of 9 rows describe very general and well-known knowledge.  Same applies for the text of the manuscript. The pioneered findings as well as recent advances in the field are not cited. The differences between responses of different sex and/or sex of the cells. The  challenges are not discussed. The different point of view is not presented. State of the art is not presented. In summary, there is no new information in the manuscript.

The friendly recommendation for the authors is to write the manuscript related to their field of expertise.

I would greatly appreciate if you will contact me if you find something in my comments is missing/unclear/incorrect.

Please notice that I do not want to review the revised version if Editor decide to give chance for the revision since my modest oppinion is that the authors are not qualify to write the review on the subject they did not published.

Good luck and all the best J

Author Response

The senior/corresponding author has large number of publications related to mitochondrial diseases in general and mitochondrial dynamics and mtDNA maintenance defects in particular. The full list can be reviewed by searching PubMed for El-Hattab AW. 

Reviewer 2 Report

Minor concerns:

- Pag 2 line 46 : authors need to explain better mitochondria metabolic adapts

- Pag 5 line 232: authors need to add more information about : new CMT2A management: 1) gene therapy   with Mfn1 adenovirus ( Yuequin Zhau 2019 JCI), and mitofusins activator (Franco A. Elife 2020, Dang X. journal experimental medicine 2020 and Dang X. journal of experimental medicine 2021)

- Improve Fig 1 , schematic representation  is not clear to understand  

Author Response

Reviewer 2

  1. Pag 2 line 46: authors need to explain better mitochondria metabolic adapts

Response: We added the following to page 2, first paragraph: Mitochondrial morphology reflects the respiratory activity of the cell. Maximum respiratory activity necessitates the fusion of the mitochondria, whereas, during cellular nutrient excess or cellular dysfunction mitochondrial fragmentation occurs. In respiratory active cells, mitochondria fuse to allow spreading of mitochondrial contents, counteract the effect of mitochondrial mutations that accumulate with aging, and optimizes mitochondrial function. Mitochondria fragment in resting cells in order to re-move damaged content by autophagy

  1. Pag 5 line 232: authors need to add more information about: new CMT2A management: 1) gene therapy with Mfn1 adenovirus (Yuequin Zhau 2019 JCI), and mitofusins activator (Franco A. Elife 2020, Dang X. journal experimental medicine 2020 and Dang X. journal of experimental medicine 2021)

Response: we expanded the discussion about mitofusin activators and added the gene therapy approach as suggested. We added the following to page 6, 3rd paragraph: Mitofusin agonists and activators are showing promising results as a therapeutic approach for CMT2A and other diseases of impaired neuronal mitochondrial dynamics (47,48). Mitofusin agonists stabilize the fusion-permissive open confirmation of endogenous normal MFN1 or MFN2. This overcomes the dominant suppression of mitochondrial fusion induced by the dysfunctional proteins and directly stimulates mitochondrial fusion in order to restore the balance between mitochondrial fission and fusion (47), (49). In mice that express human MFN2 T105M, intermittent activation of mitofusin using MiM111, which is a metabolically stable mitofusin activator with good nervous system bioavailability, normalized CMT2A neuromuscular dysfunction secondary to accelerated primary axonal outgrowth and greater post-axotomy regrowth (48). Gene therapy is another promising approach. It has been shown that in vivo augmentation of MFN1 in the central nervous system of mice, by using a transgenic approach, rescued all phenotypes in mutant MFN2-expressing mice (50).

  1. Improve Fig 1, schematic representation is not clear to understand

Response: Figure 1 has been modified 

Reviewer 3 Report

The description of disorders is fine and well written. However, the Introduction and the parts describing the mechanisms and functions are rough with some inaccurate information. The authors should cite the original papers, not the papers that cite the original papers.

1.     Line 17: “coding” needs to be changed to “coding for” or “encoding”

2.     Line 93: The OMA1, not Yme1L, is the stress-activated protease that cleaves L-OPA1.

S-OPA1 is not an ‘inactive isoform’. See these papers

doi: 10.1074/jbc.M116.762567

doi: 10.1016/j.celrep.2017.05.073

3.     Line 103: This appears to be a simple speculation based on morphological observation. Is there direct evidence for FBXL4 interacting with fusion machinery? If not, it cannot be said to be a fusion protein.

4.     Lines 105 – 120, and lines 134 – 144: It is recommended not to simply list the findings/theories/speculations. It is necessary to describe these in a coherent manner and provide a conclusive sentence at the end of paragraph.

5.     Line 125: Fis1 as a DNM1L receptor is debated and it may not be a Drp1 receptor after all.  MiD49/51 are the legitimate receptors along with Mff.

6.     Line 133: How does ubiquitination stabilize DNM1L?

7.     Line 146: Figure 1 needs more attention regarding the protein locations. Mitochondrial membranes and IMS need to be enlarged to accommodate the correct locations of the proteins

Author Response

Reviewer 3

The description of disorders is fine and well written. However, the Introduction and the parts describing the mechanisms and functions are rough with some inaccurate information. The authors should cite the original papers, not the papers that cite the original papers.

Response: we added more original references and reviewed and edited the paper as suggested by the reviewer below.

  1. Line 17: “coding” needs to be changed to “coding for” or “encoding”

Response: this error has been corrected

  1. Line 93: The OMA1, not Yme1L, is the stress-activated protease that cleaves L-OPA1. S-OPA1 is not an ‘inactive isoform’. See these papers

doi: 10.1074/jbc.M116.762567

doi: 10.1016/j.celrep.2017.05.073

Response: We thank the reviewers to raise these points. We modified the first paragraph in page 3 as following: When the transmembrane potential across MIM is intact, long OPA1 (L-OPA1) isoforms carry out MIM fusion. When the potential is lost, L-OPA1 is cleaved to short (S-OPA1) by stress sensitive IMS metalloendopeptidase OMA1 (OMA1; encoded by the OMA1 gene) (19) with increased S-OPA1 inhibiting the fusion process and promoting mitochondrial fragmentation (20). The ATP-dependent zinc metalloprotease YME1L1 (YME1L1; encoded by YME1L1) catalyses the degradation of OMA1 in response to membrane depolarization (21).

  1. Line 103: This appears to be a simple speculation based on morphological observation. Is there direct evidence for FBXL4 interacting with fusion machinery? If not, it cannot be said to be a fusion protein.

Response: We agree with the reviewer that we do not have evidence that FBXL4 is a fusion protein; however, it plays role in mitochondrial fusion and this can be mediated via the interaction with mitochondrial fusion proteins. We modified the second paragraph of page 3 as following: FBXL4 may play a role in mitochondrial fusion via interacting with and regulating mitochondrial fusion proteins (22).  

  1. Lines 105 – 120, and lines 134 – 144: It is recommended not to simply list the findings/theories/speculations. It is necessary to describe these in a coherent manner and provide a conclusive sentence at the end of paragraph.

Response: We modified the third paragraph of page 3 as following: Mitochondrial fusion allows the cells to build interconnected mitochondrial networks by producing tubular or elongated mitochondria. These networks act as united systems promoting oxidative phosphorylation and leading to efficient dissipation of energy in the cells. Therefore, these networks are frequently found in metabolically active cells (23) Mitochondrial fusion also enables content mixing within the mitochondrial population, hence avoiding permanent loss of essential components and unifying the mitochondrial compartment which optimizes mitochondrial function (24). This mixing also allows the redistribution of mtDNA between damaged and healthy mitochondria which enables human cells to tolerate high levels of pathogenic mtDNA and prevents mitochondrial elimination via mitophagy (25), (26). Hence mitochondrial fusion is an essential regulatory process for optimizing mitochondrial function and enhancing mitochondrial integrity by allowing component sharing. 

We also modified the last paragraph in page 3 as following: Mitochondrial fission creates new mitochondria which is crucial for rapidly dividing and growing cells to populate them with adequate numbers of mitochondria (30). Mitochondrial fission also allows the redistribution of mitochondria. Through the formation of smaller mitochondria, mitochondrial fission allows a more efficient redistribution of these smaller sized organelles to the energy demanding regions. Mitochondrial fission also contributes to quality control by enabling the removal of damaged mitochondria as it can isolate impaired mitochondria to be eliminated by mitophagy which maintains mitochondrial homeostasis (31). Therefore, mitochondrial fission is essential for mitochondrial distribution and homoeostasis.

  1. Line 125: Fis1 as a DNM1L receptor is debated and it may not be a Drp1 receptor after all. MiD49/51 are the legitimate receptors along with Mff.

Response: we modified the 4th paragraph of page 3 as following: DNM1L acts on several MOM receptors including mitochondrial fission factor (MFF; encoded by MFF), mitochondrial dynamics protein 49 (MID49), and mitochondrial dynamics protein 51 (27).

  1. Line 133: How does ubiquitination stabilize DNM1L?

Response: we apologize for this mistake. Ubiquitination leads to the degradation of DNM1L not stabilization. Therefore, we modified the 4th paragraph of page 3 to: Posttranslational phosphorylation, SUMOylation, and ubiquitination regulate DNM1L on the mitochondria (29).

  1. Line 146: Figure 1 needs more attention regarding the protein locations. Mitochondrial membranes and IMS need to be enlarged to accommodate the correct locations of the proteins

Response: Figure 1 has been modified

Round 2

Reviewer 3 Report

The authors' responses are adequate.

One clarification is needed: lines 103 - 105. This happens only when ATP is available. If ATP is depleted with the mitochondrial depolarization, YME1L is inhibited, which stabilizes OMA1 to cleave L-OPA1 to S-OPA1.